# *Mycobacterium avium* subsp. *paratuberculosis* Candidate Vaccine Strains Are Pro-apoptotic in RAW 264.7 Murine Macrophages

**DOI:** 10.3390/vaccines11061085

**Published:** 2023-06-10

**Authors:** Raul G. Barletta, John P. Bannantine, Judith R. Stabel, Ezhumalai Muthukrishnan, Dirk K. Anderson, Enakshy Dutta, Vamsi Manthena, Mostafa Hanafy, Denise K. Zinniel

**Affiliations:** 1School of Veterinary Medicine and Biomedical Sciences, University of Nebraska, Lincoln, NE 68583, USA; ezhumalai085@gmail.com (E.M.); mhanafy2@unl.edu (M.H.); dzinniel2@unl.edu (D.K.Z.); 2United States Department of Agriculture-Agricultural Research Service, National Animal Disease Center, Ames, IA 50010, USA; john.bannantine@usda.gov (J.P.B.); judy.stabel@usda.gov (J.R.S.); 3Nebraska Center for Biotechnology, Flow Cytometry Core Facility, University of Nebraska, Lincoln, NE 68588, USA; dirk.anderson@unl.edu; 4Department of Statistics, University of Nebraska, Lincoln, NE 68583, USA; enakshy.dutta@huskers.unl.edu (E.D.); vamsi.manthena@gmail.com (V.M.); 5Department of Microbiology and Immunology, Faculty of Veterinary Medicine, Cairo University, Giza 12211, Egypt

**Keywords:** *Mycobacterium avium* subsp. *paratuberculosis*, macrophages, virulence, apoptosis, necrosis, vaccine, bovine

## Abstract

*Mycobacterium avium* subsp. *paratuberculosis* (MAP) is the etiological agent of Johne’s disease, a severe gastroenteritis of ruminants. This study developed a model cell culture system to rapidly screen MAP mutants with vaccine potential for apoptosis. Two wild-type strains, a transposon mutant, and two deletion mutant MAP strains (MOI of 10 with 1.2 × 10^6^ CFU) were tested in murine RAW 264.7 macrophages to determine if they induce apoptosis and/or necrosis. Both deletion mutants were previously shown to be attenuated and immunogenic in primary bovine macrophages. All strains had similar growth rates, but cell morphology indicated that both deletion mutants were elongated with cell wall bulging. Cell death kinetics were followed by a real-time cellular assay to measure luminescence (apoptosis) and fluorescence (necrosis). A 6 h infection period was the appropriate time to assess apoptosis that was followed by secondary necrosis. Apoptosis was also quantified via DAPI-stained nuclear morphology and validated via flow cytometry. The combined analysis confirmed the hypothesis that candidate vaccine deletion mutants are pro-apoptotic in RAW 264.7 cells. In conclusion, the increased apoptosis seen in the deletion mutants correlates with the attenuated phenotype and immunogenicity observed in bovine macrophages, a property associated with good vaccine candidates.

## 1. Introduction

*Mycobacterium avium* subsp. *paratuberculosis* (MAP) is the causative agent of Johne’s disease (JD). This disease has a significant economic impact on the livestock industry worldwide. Common disease symptoms are chronic diarrhea, malabsorption, body weakness, and malnutrition, characterized by the development of lesions in the intestinal mucosa of ruminants [1]. JD results in decreased health status, which necessitates a premature elimination of diseased animals, leading to major losses. This is of significant concern for the dairy industry since there is a decrease in milk production [2]. Herd management is usually accomplished by the test and cull strategy, but vaccination would be the most cost-effective method to control paratuberculosis [3], decreasing the number of clinical cases and improving milk production [4,5]. In addition, MAP has been associated with Crohn’s disease and other autoimmune disorders. This hypothesis has been strengthened by the cross-reactivity between MAP and human proteins, suggesting a mechanism of molecular mimicry [6].

Cell death processes are characterized by their morphological features [7]. Apoptosis has plasma membrane blisters, decreased cellular and nuclear volume, nuclear fragmentation, and slight changes in cytoplasmic organelles. In contrast, necrosis exhibits plasma membrane rupturing, cytoplasmic inflammation with enlarged organelles, and modest chromatin condensation. Cell death mechanisms and their regulation play important roles in cellular homeostasis and host defenses against bacteria, viruses, and other infectious agents [8]. MAP has the ability to survive and replicate inside host macrophages by preventing normal phagosome maturation [9,10]. In general, intracellular bacterial pathogens inhibit apoptosis as a survival strategy allowing replication and invasion [11]. Cell death studies on mycobacterial pathogens have focused on *Mycobacterium tuberculosis* (MTB). Instead of apoptosis, virulent wild-type MTB induces macrophage and monocyte necrosis, which results in host cell lysis, leading to tissue damage, replication, and spread [12]. In contrast, live-attenuated MTB vaccine candidates favor apoptosis. Similar findings were observed for MAP indicating that the wild-type strains inhibited apoptosis in monocyte-derived macrophages and attenuated mutants were pro-apoptotic [13]. Interestingly, in vivo studies in MAP concluded that apoptosis is greater when macrophages contain higher numbers of bacilli, but the significance of these results has not been elucidated [14]. This may represent a later stage in JD as MAP bacilli were found in foamy macrophages, and this process would not apply to tissue culture experiments assessing the initial interactions.

The preponderance of the evidence supports the concept that apoptosis is inhibited by wild-type strains, while attenuated mutants induce apoptosis. Nonetheless, a few contradictory results were described according to which wild-type MTB induced apoptosis in murine macrophages and C57BL/6 mice, while *Mycobacterium bovis* BCG and the attenuated MTB *phoP* mutant failed to promote apoptosis [15]. Likewise, wild-type *Mycobacterium avium* subsp. *hominissuis* was shown to induce apoptosis in the human monocytic cell line THP-1 [16]. These discrepancies may be due to the use of different mycobacterial species, primary macrophages versus cell lines, the multiplicity of infection (MOI), assay endpoints, and methodology.

Bovine monocyte-derived primary macrophages are preferred for studying host–pathogen interactions in JD. However, these cells pose problems for large screening assays, as several animals will need to be used for blood collection, thus increasing variability and labor. Since the goal of this study was to develop a method to screen a large MAP mutant collection for apoptotic properties, a cell line would be desirable to rapidly perform this test. Albeit a bovine macrophage cell line (BoMac) was developed, it does not reflect MAP interactions with primary macrophages due to poor phagocytosis and a decreased ability to sustain intracellular MAP replication [17]. In contrast, RAW 264.7 macrophages have been shown to provide a good model for bovine macrophages regarding immunogenicity and apoptosis [18,19]. Thus, we chose to use this well-established murine macrophage cell line derived from BALB/c mice to compare cell death processes elicited by MAP strains. Our results indicated that BALB/c mice are more susceptible to MAP infection [20]. Moreover, a previous study successfully utilized BALB/c mice to screen a large transposon mutant library [21]. Some of these mutants were characterized as immunogenic and attenuated in ruminants validating the BALB/c murine model [22,23]. Based on all of these studies, BALB/c mice and their derived cell lines provide a good model that can be correlated with bovine experiments.

Previous research on mycobacterial pathogens has led to the consensus that live-attenuated vaccines are the best approach for vaccination [24]. Our laboratory has established a large MAP collection comprising more than a million mutants, including Tn*5367* and *Himar1* transposon insertion mutant banks [25,26,27]. In addition, two deletion mutants (DMAP52 and DMAP56) in ORFs MAP_1152 and MAP_1156 were isolated. MAP_1152 encodes a PPE protein, and a gene in the same cluster, MAP_1156, encodes a diacylglycerol acyltransferase involved in triglyceride metabolism [28]. Both mutants were determined to be attenuated in bovine macrophages, as shown by reduced bacterial burdens in comparison to the wild-type strain upon incubation for 48 h [29]. Moreover, in this study, these mutants were tested for immunogenicity by performing cell proliferation assays. Upon 72 h incubation in peripheral blood mononuclear cells, the percent of activated cells for CD4, CD8, and gamma-delta T cells were the same for all wild-type and mutant strains and greater than the nonstimulated response.

We aimed to develop RAW 264.7 macrophages as a model cell culture system to rapidly screen this collection for apoptotic properties that could then be tested for vaccine potential. We hypothesize that attenuation and immunogenicity in primary bovine macrophages correlate with apoptosis in this murine cell line. To test cell death processes, a real-time assay, DAPI staining, and flow cytometry were performed to detect apoptosis and/or necrosis. We found that the results from apoptosis in RAW 264.7 using all methods correlated well with attenuation in primary bovine macrophages. The deletion mutants DMAP52 and DMAP56 displayed both attenuation in bovine macrophages and pro-apoptotic properties in RAW 264.7, making them excellent vaccine candidates against JD.

## 2. Materials and Methods

### 2.1. Bacterial Strains, Growth Curves, and Cell Morphology

The MAP strains utilized in this study include two wild-type strains: University of Nebraska-Lincoln (UNL), Lincoln, NE, USA strain K-10 [30] and National Animal Disease Center (NADC) K-10, which is an isolate of K-10 recently passaged in calves received from the NADC in Ames, IA, USA [31]. In addition, three mutants derived from UNL K-10 were also tested: 4H2, DMAP52, and DMAP56. Strain 4H2 was a kanamycin-resistant Tn*5367* transposon insertion mutant with an insertion in the intergenic region between MAP1150c and MAP1151c that is upstream from the MAP_1152-MAP_1156 cluster [25]. DMAP52 (MAPΔMAP_1152) and DMAP56 (MAPΔMAP_1156) are hygromycin-resistant deletion mutants generated via allelic exchange [29] using the conditionally replicating phagemid phAE87 [32]. MAP strains were grown to the late exponential phase, to an optical density at 600 nm (OD_600_), between 1.0 and 1.5 at 37 °C in complete Difco^™^ Middlebrook 7H9 base broth (Becton, Dickinson and Company; Sparks, MD, USA), adjusted to pH 5.9, and supplemented with BBL^™^ oleic acid/albumin/dextrose complex (Becton, Dickinson and Company; Sparks, MD, USA), ferric mycobactin J (1.0 μg/mL), and Tween^®^ 80 (Thermo Fisher Scientific; Waltham, MA, USA) (MOADC), as previously described [30]. The growth curves were determined as previously described [25]. Briefly, MAP cultures with an initial OD_600_ of 0.05 were grown standing at 37 °C in T75 vented flasks over 28 days, and at various times, OD_600_ readings were carried out, and colony forming units (CFUs) were determined on MOADC agar after 5 weeks of incubation. The data for the transposon mutant 4H2 were taken from Rathnaiah et al. (2014) [25].

For cell morphology, MAP strains were grown to an OD_600_ of about 1.0 in MOADC. Bacteria were harvested and washed twice with PBS with 0.05% Tween^®^ 80 and resuspended in double distilled water to a final concentration of 8.0 × 10^8^ CFU/mL. Bacterial cells were transferred to a 300 mesh Formvar Carbon Type B copper grid (Ted Pella, Inc.; Redding, CA, USA) using a nebulizer and fixed with 2% phosphotungstic acid pH 6.5 and 0.01% bovine serum albumin. Grids were dried overnight and visualized using a Hitachi High-Tech America, Inc., H7500 transmission electron microscope (TEM; Schaumburg, IL, USA) with a magnification range of 3000× to 10,000× in the HC mode at 80 KV.

### 2.2. RAW 264.7 Macrophage Cells and MAP Infection Protocol

Murine RAW 264.7 macrophage cells (ATCC™; Manassas, VA, USA) from BALB/c male mice were grown in Dulbecco’s modified Eagle’s medium (DMEM) (ATCC™; Manassas, VA, USA) supplemented with 10% fetal bovine serum and 50 µg/mL of both penicillin and streptomycin to keep the sterility of the media, incubating at 37 °C in 5% CO_2_. In addition, streptomycin has the ability to kill extracellular MAP [33]. These conditions were maintained throughout the course of all experiments. MAP cultures were grown to an OD_600_ between 0.4 and 0.6 in MOADC. Each strain (1.0 mL) was centrifuged at 3300× *g* for 10 min and resuspended in a 1.5 mL basal uptake buffer (BUB). Bacteria were passed 10–15 times through a syringe with a 26.5-gauge needle to remove clumps. Finally, these bacterial cells were added to 3.5 mL BUB, mixed, and used to infect macrophage cells at an MOI of 10 [34].

### 2.3. Real-Time Apoptosis and Necrosis Assays

RAW 264.7 cells were seeded with a density of 1.0 × 10^4^ cells/well on 96-well white plates (96F Nunclon^™^ Delta White Microwell SI; Thermo Fisher Scientific; Waltham, MA, USA) overnight (ca. 18 h) to a final density of 6.0 × 10^4^ cells/well and infected with MAP at an MOI of 10. Following the protocol provided by the Real-time-Glo™ Annexin V Apoptosis and Necrosis Assay (Promega; Madison, WI, USA), the proprietary detection reagents (Annexin V NanoBiT^®^ Substrate, Necrosis Detection Reagent, Annexin V-LgBiT, and Annexin V-SmBiT) were added, and samples were incubated for 1, 3, 6, 12, 24, and 48 h. Data were collected using a Molecular Devices SpectraMax^®^ Paradigm^®^ Multi-Mode Detection Platform (Sunnyvale, CA, USA) at 578 nm for luminescence (apoptosis) and excitation at 485 nm/emission at 535 nm for fluorescence (necrosis).

### 2.4. Apoptosis Assay via DAPI Staining

The percent of apoptosis was quantified via characteristic nuclear morphology and visualized via treatment with the fluorescent DNA-binding dye DAPI (4′,6-diamidino-2-phenylindole) (MilliporeSigma™ Calbiochem™; Burlington, MA, USA). RAW 264.7 cells were seeded into 24-well plates at a density of 5.0 × 10^4^ cells/well, grown overnight to a final density of 2.0 × 10^5^ macrophage cells/well (ca. 18 h), and infected with MAP (MOI of 10). After a 6 h incubation, cells were washed with PBS at least two times and stained with 0.005 μg of DAPI for 15–30 min at 37 °C. The percent of apoptotic nuclei was determined using a fluorescence microscope (Nikon ECLIPSE TE2000-U; Melville, NY, USA), with a magnification of 200×.

### 2.5. Apoptosis Assay via Flow Cytometry

RAW 264.7 cells (3.0 × 10^5^ cells/well) were seeded into 6-well plates, grown overnight (ca. 18 h) to a final density of 1.2 × 10^6^ cells/well, and infected with MAP at an MOI of 10. After an incubation of 3–5 h, DMEM was replaced, and cells were further incubated for a total of 6 h. Following the recommended procedures for the FITC Annexin V Apoptosis Detection Kit I (BD Pharmingen^™^; Franklin Lakes, NJ, USA), the infected cells were washed with PBS and resuspended in 1.0 mL of 1X Annexin V binding buffer. An aliquot of 100 µL was mixed by gently pipetting up and down with 5 µL each of the florescent dyes FITC Annexin V and 7-amino-actinomycin (7-AAD) (Cayman Chemical; Ann Arbor, MI, USA), incubating in a dark place for 15 min at RT (25 °C). Finally, 400 µL of 1X Annexin V binding buffer was added to each tube, and the fluorescence output was analyzed via flow cytometry within one hour using a Beckman Coulter^®^ CytoFLEX LX instrument (Brea, CA, USA). Fluorescence-activated cell sorting (FACS) dot plots were analyzed using the FlowJo™ software version 10.8.1 (FlowJo™, LLC; Ashland, OR, USA).

### 2.6. Statistics

Statistical analysis was performed using SAS version 9.4, 2012 (SAS Institute, Inc.; Cary, NC, USA). All experiments were conducted with at least three biological replicates each and with three technical replicates. Growth curve data were converted to log_10_, while the natural log was used for the cell morphology, real-time apoptosis and necrosis, DAPI staining, and flow cytometry experiments. SAS Proc Mixed was applied for the growth curves, while Proc Glimmix using the Tukey–Kramer HSD test was employed for all other experiments. As needed, data outliers were removed and replaced with the average value for each strain and checked for normality. Pairwise determinations for statistical significance were calculated: * *p* ≤ 0.05, ** *p* ≤ 0.01, and *** *p* ≤ 0.001.

## 3. Results

### 3.1. Growth Curves and Cell Morphology

To determine whether wild-type and MAP mutants have similar in vitro growth rates in broth culture, which is a desirable property for vaccine development, OD_600_ values and agar colony count data (CFU/mL) were collected. As shown in Figure 1A, the growth rate based on the OD_600_ slope from Day 0 to 14 for each strain was not significantly different compared with UNL K-10 except for 4H2 (*p* = 0.0341). In addition, there were significant differences observed when comparing NADC K-10 to 4H2 (*p* = 0.0015) and DMAP56 (*p* = 0.0038). For CFU/mL vs. time, no significant differences were found for the corresponding slopes (Figure 1B). Thus, the strains grew at approximately the same rate. Surprisingly although all strains were inoculated to an initial OD_600_ of 0.05, the DMAP52 and DMAP56 CFU/mL were 10- and 18-fold lower than UNL K-10, respectively. We utilized this relationship to calculate the MOI for subsequent experiments. Based on these results, we hypothesize that the discrepancies between OD_600_ and CFU/mL were due to differences in cell morphology [35].

Therefore, TEM micrographs were collected (Figure 2A) from copper grids containing MAP fixed with phosphotungstic acid and bovine serum albumin. The strains UNL K-10 and NADC K-10 had a similar shape and morphology, as indicated by their average lengths (1.31 to 1.44 µm) and widths (0.46 to 0.47 µm). The transposon 4H2 mutant (1.58 × 0.48 µm) was significantly different compared with NADC K-10 for length (Figure 2B; *p* = 0.0007). In contrast, DMAP52 (1.99 × 0.51 µm) and DMAP56 (1.93 × 0.54 µm) had substantial elongation compared with the wild types and 4H2 in length (Figure 2B, *p* < 0.001). Regarding cell width, the deletion mutants had significant bulging of the cell wall (Figure 2C; DMAP52 vs. UNL K-10 was *p* = 0.0242, NADC K-10 was *p* = 0.0008, and 4H2 was *p* = 0.0209, while DMAP56 were all *p* < 0.0001).

### 3.2. Real-Time Apoptosis and Necrosis Assays

The Real-time-Glo™ Annexin V Apoptosis and Necrosis Assay can determine both apoptosis (luminescence) and necrosis (fluorescence) cell death processes in the same experiment using MAP-infected RAW 264.7 cells over time (1 to 48 h). The apoptosis data (Figure 3A) indicate that as incubation time progressed, the levels for the uninfected cell control, NADC K-10, UNL K-10, and 4H2 increased from 1 to 3 h (*p* ≤ 0.0002). For these samples, after the 3 h peak, the luminescence decreased significantly at 24 and 48 h (*p* ≤ 0.0001). For DMAP52 and DMAP56 from 1 h, apoptosis significantly increased, reaching the maximum at 6 h (*p* < 0.0001). Thereafter, significant decreases were observed at 24 and 48 h (*p* < 0.0001). When comparing NADC K-10 (*p* = 0.0034), UNL K-10 (*p* = 0.0031), and 4H2 (*p* = 0.0006) to the control, noteworthy changes were only found at 48 h. For the deletion mutants, significant RLU differences to the control were found at 6 h (DMAP52, *p* = 0.0236; DMAP56, *p* = 0.0304), 12 h (*p* < 0.0001), 24 h (DMAP52, *p* = 0.0013; DMAP56, *p* = 0.0007), and 48 h (*p* < 0.0001). At 12 h, there were significant apoptosis differences comparing NADC K-10 vs. DMAP52/DMAP56 (*p* = 0.0004), UNL K-10 vs. DMAP52 (*p* = 0.0049)/DMAP56 (*p* = 0.005), and 4H2 vs. DMAP52/DMAP56 (*p* = 0.0001). At 24 h, there were significant luminescence differences between DMAP52 and NADC K-10 (*p* = 0.0252) and 4H2 (*p* = 0.0478), as well as between DMAP56 and NADC K-10 (*p* = 0.0153), UNL K-10 (*p* = 0.0383), and 4H2 (*p* = 0.03).

The fluorescence output (Figure 3B) indicates that necrosis increased over all time points except for a nonsignificant decrease for the uninfected control at 24 h. The earliest significant RFU differences observed after 1 h was at 6 h for DMAP52 (*p* = 0.0224) and DMAP56 (*p* = 0.0311), 12 h for UNL K-10 (*p* = 0.0449) and 4H2 (*p* = 0.0271), and 24 h for the control (*p* = 0.0281) and NADC K-10 (*p* = 0.013). For the deletion mutants, there was a large fluorescence increase between 3 and 12 h (*p* < 0.01) and 6 and 12 h (*p* < 0.0001). The only significant RFU difference observed for each time point was at 24 h for DMAP52 (*p* = 0.0333) and DMAP56 (*p* = 0.0148). The differences between DMAP56 and the control (12 h), NADC K-10 (24 and 48 h), and UNL K-10 (24 and 48 h) were close to reaching significance. These results are consistent with an apoptotic process, indicated by an initial luminesce increase followed by an output decline (Figure 3A), while necrosis fluorescence increased throughout the duration of the experiment (Figure 3B). Thus, we interpret these results as macrophage cells undergoing an apoptotic process followed by secondary necrosis.

### 3.3. Apoptosis Assay via DAPI Staining

The real-time cellular assays above identified the 6 h incubation period as the apoptotic peak for the deletion mutants and therefore an appropriate time to detect apoptosis via DAPI staining upon infecting RAW 264.7 cells with MAP. As shown in Figure 4A, the fold changes in the percent of apoptotic nuclei compared with the uninfected cells (control) for NADC K-10 (4.9), UNL K-10 (5.7), and 4H2 (6.0) were similar, while there was a substantial increase for DMAP52 (48.3) and DMAP56 (60.0). There were no significant differences between the wild types and the transposon mutant. However, DMAP52 and DMAP56 showed a significant increase in apoptotic nuclei (*p* < 0.0001) with respect to MAP strains NADC K-10, UNL K-10, and 4H2. Figure 4B displays the corresponding phase contrast and fluorescent micrographs for all RAW 264.7 macrophage samples, which illustrates the large number of DAPI-stained cells observed for the deletion mutant strains.

### 3.4. Apoptosis Assay via Flow Cytometry

The well-established flow cytometry method was employed with the FITC Annexin V Apoptosis Detection Kit I. RAW 264.7 cells infected with MAP for 6 h and labeled with FITC Annexin V and 7-AAD dyes were divided into four stages based on the intensity of the fluorescent signal. The gate positions were determined using stained and single-color controls. Figure 5A displays representative FACS dot plots for uninfected RAW 264.7 cells and those infected with each MAP strain in various stages: live (bottom left), early apoptosis (bottom right), late apoptosis (top right), and necrosis (top left). In general, as shown in Figure 5B, the MAP strains could be divided into two groups with low (wild types and transposon mutant) and high (deletion mutants) levels of early (diagonal lines), late (solid), and total (early plus late) apoptosis. Therefore, it is not surprising that significant differences were identified for DMAP52 compared with UNL K-10 (early, *p* = 0.0054; total, *p* = 0.0257) and 4H2 (early, *p* = 0.0275). For DMAP56, noteworthy changes were with NADC K-10 (late, *p* = 0.0015; total, *p* = 0.0034), UNL K-10 (early, *p* = 0.0071; late, *p* = 0.0019; total, *p* = 0.0007) and 4H2 (early, *p* = 0.0332; late, *p* = 0.0089; total, *p* = 0.0083). These flow cytometry results are consistent with the real-time and DAPI staining data indicating that DMAP52 and DMAP56 are pro-apoptotic.

## 4. Discussion

JD is a chronic inflammatory disease caused by MAP, with a high worldwide prevalence. In the United States alone, the dairy herd-level prevalence of MAP is approaching 100% [36]. This high prevalence is not surprising since producers unknowingly purchase animals that are infected. Control measures for JD may include vaccination, annual herd screening using diagnostic tests, and improved herd management based on a producer’s financial resources, facilities, and operation [37]. Vaccination is one of the most cost-effective disease control measures [38]. Unfortunately, current JD vaccines have limited efficacy. The major concern with killed and candidate live-attenuated vaccines against JD is the interference with diagnostic tests yielding false-positive results [39,40]. However, we can still address antigenic cross-reactivity with *M. bovis* and MAP natural infection by developing a formulation to distinguish animals vaccinated from those infected [41]. In previous studies, we discovered that two novel MAP antigens (MAP_1152 and MAP_1156) were able to differentiate the serum samples of uninfected cattle from those of MAP-infected cattle [28]. These results suggest that cattle vaccinated with MAP deletion mutants in the corresponding genes would yield a negative humoral test, while a positive result may be obtained for naturally infected cattle.

Previous studies indicated that the C57BL/6 mouse model does not reflect JD in ruminants [42,43]. Moreover, this mouse model showed a decrease in MAP CFUs over time in the liver compared with the less immunocompetent BALB/c mice, which is an indication that C57BL/6 mice are somewhat resistant to infection [20]. Thus, to develop a model cell culture system, RAW 264.7 macrophage cells derived from BALB/c mice were utilized. Studies with MAP using primary bovine macrophages, separating bystander from infected cells, showed that virulent strains inhibited apoptosis, while vaccine-candidate strains induced apoptotic changes [13,44]. In our experiments, a whole-tissue culture-based method was employed without the separation of bystander and infected cells. The deletion mutants showed significantly increased levels of apoptosis compared with the wild-type strains and 4H2. However, the inhibition of apoptosis was not observed for either of the wild-type strains in our experiments, similar to the previous study cited above [13]. The use of a murine macrophage cell line instead of monocyte-derived bovine macrophages is a potential reason for not observing apoptosis inhibition by wild-type cells, although the omission of cell separation is the most likely explanation for this effect. Moreover, apoptotic processes in bovine macrophages and RAW 264.7 cells are very similar, as the same effector (e.g., the programmed cell death protein-4) and regulatory (e.g., miR-150) mechanisms are present in both bovine and mouse macrophages [19]. Thus, the RAW 264.7 cell line is an excellent model to study MAP apoptosis, as our results demonstrate that MAP strain pro-apoptotic properties can be assessed in this cell line.

In our studies, growth analysis indicated that all MAP strains grew in broth cultures at approximately the same rate, which is a desirable property for vaccine candidates. Interestingly, TEM micrographs indicated that the deletion mutants in MAP_1152 and MAP_1156 displayed an increased length and width. Both proteins are localized to the cell surface or membrane, where such deletions may be expected to result in morphological changes. In contrast, these morphological changes were not observed for MAP 4H2, a transposon colony morphology mutant whose insertion was mapped to an intergenic region upstream from the MAP_1152–MAP_1156 gene cluster [25]. Although the deletion mutants were longer than the corresponding wild type, their size alone would be unlikely to change their uptake by macrophages, a general property that has been described in the literature [45,46].

Three methods with different dyes were performed to detect apoptosis and/or necrosis: a real-time assay, DAPI staining, and flow cytometry. A real-time assay to determine the kinetics of cell death processes was applied for the first time in MAP. Apoptosis causes the disruption of the cell membrane, resulting in the exposure of phosphatidylserine to the cell membrane exterior, thus facilitating the two complementary Annexin V luciferase fusion proteins to bind and form the NanoBIT^®^ luciferase complex [47,48,49]. The luciferase generates a luminescence output (RLU) in the presence of a substrate provided using the detection reagent. Upon complete loss of inner and outer leaflet membrane integrity, the necrosis cell-impermeable cyanine dye enters the cells, binds DNA, and generates a strong fluorescence signal (RFU) associated with necrosis [50]. Based on these results, apoptosis peaked for the deletion mutants at 6 h, and thus the simple DAPI staining method under these conditions preferentially allowed for the staining of apoptotic cell nuclei undergoing DNA fragmentation. This dye can penetrate less efficiently through an intact cell membrane, making it ideal to test apoptotic and necrotic macrophages with compromised cell membranes [51]. DAPI strongly binds to adenine + thymine (A + T)-rich DNA sequences, resulting in nuclear fluorescence. Consistent with the real-time assay, DAPI staining demonstrated that the deletion mutants had a higher percentage of apoptotic nuclei. Flow cytometry is the most widely used method to study cell death processes; thus, experiments were carried out to confirm the results obtained with those of the previous two methods [52]. The FlowJo™ software (FlowJo™, LLC; Ashland, OR, USA) was used to carefully gate the MAP cell populations. This is important as flow cytometry is negatively influenced by the nonspecific binding of antibodies, autofluorescence, and/or cell clumping. In general, the deletion mutants were also shown to be pro-apoptotic either in early, late, or total apoptosis compared with the parent wild-type strain. Interestingly, this was not the case for 4H2, which was previously shown to be attenuated in primary bovine macrophages, but its immunogenicity was not assessed [25]. Moreover, the transposon insertion in 4H2 is in an intergenic region and the presence of the Tn*5367* transposase may cause instability. Thus, we no longer consider this strain as an appropriate vaccine candidate.

## 5. Conclusions

The combined analysis of all methods confirmed the hypothesis that candidate vaccine strains DMAP52 and DMAP56 are pro-apoptotic in RAW 264.7 murine macrophages. None of these tests identified any significant differences between these two deletion mutants. The real-time assay indicated the early induction of apoptosis for all MAP strains, followed by secondary necrosis. DAPI staining was the quickest method to determine apoptosis that can be used to screen large mutant collections. Moreover, we determined that attenuation and immunogenicity in bovine macrophages could be well correlated with apoptosis in RAW 264.7 provided that appropriate kinetic measurements are performed to choose the appropriate post-infection times to assess parameters. In addition, apoptosis always preceded the necrotic process that occurred at later time points. Overall, DMAP52 and DMAP56 were found to be excellent vaccine candidates against JD. In future experiments, we plan to evaluate these deletion mutants in calves.

## Figures and Tables

**Figure 1 vaccines-11-01085-f001:**
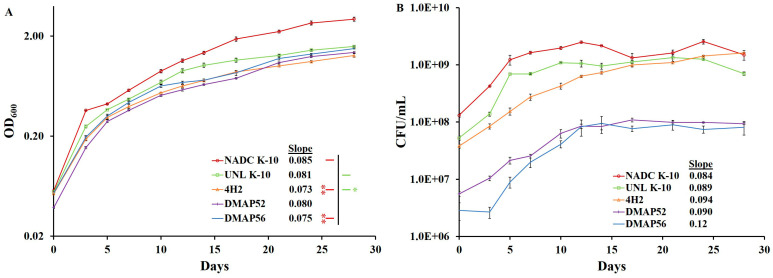
Determination of growth curves for MAP strains. Bacilli were inoculated into MOADC media at an initial OD_600_ of about 0.05. Cultures were incubated standing at 37 °C for 28 days: (**A**) OD_600_ readings were carried out at various time points, and (**B**) the corresponding CFU counts were determined; both are in the log_10_ scale. The slopes for each strain on Days 0 to 14, as well as the statistical significance of pairwise determinations, are indicated as * *p* ≤ 0.05 and ** *p* ≤ 0.01. Results represent means (*n* = 3) ± standard errors of the mean. A legend displays the correlation between each strain and the color used.

**Figure 2 vaccines-11-01085-f002:**
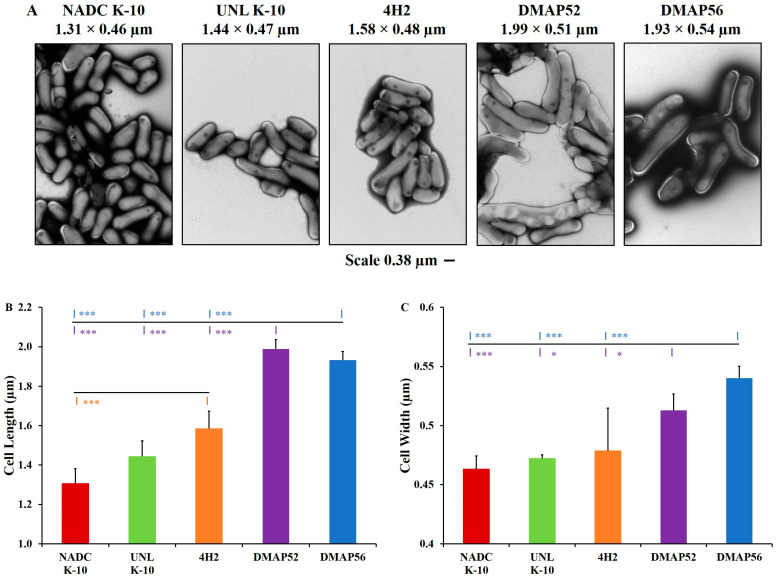
Determination of cell morphology using transmission electron microscopy: (**A**) micrographs of MAP strains with a reference bar of 0.380 µm; (**B**) strain 4H2 (orange tick mark) was significantly longer than NADC K-10. DMAP52 (purple tick marks) and DMAP56 (blue tick marks) showed significantly increased length compared with both wild types and 4H2; (**C**) the deletion mutants also had significantly increased width compared with these strains. The asterisk color represents the strain to which pairwise comparisons were statistically significant: * *p* ≤ 0.05 and *** *p* ≤ 0.001. Results represent means (*n* = 3) ± standard errors of the mean.

**Figure 3 vaccines-11-01085-f003:**
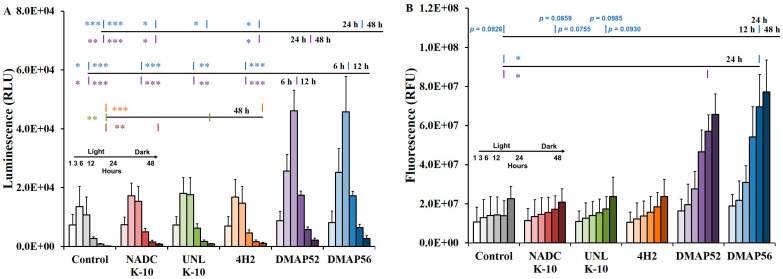
RAW 264.7 uninfected control (gray) and MAP-infected cells were assayed for real-time apoptosis and necrosis. Measurements for the various time points from 1 to 48 h are indicated by a light-to-dark color gradient: (**A**) apoptosis (RLU) was determined as a luminescence output. Pairwise comparisons are shown for DMAP56 (top) and DMAP52 (bottom) vs. the control, wild types, and 4H2 for 24/48 h (upper line) and 6/12 h (middle line). The lower line displays pairwise comparisons between 4H2 (top), UNL K-10 (inline), and NADC K-10 (bottom) vs. the uninfected cells for 48 h. The asterisk color represents the strain to which pairwise comparisons were statistically significant: * *p* ≤ 0.05, ** *p* ≤ 0.01, and *** *p* ≤ 0.001; (**B**) secondary necrosis (RFU) was recorded as a fluorescence output. The upper line has pairwise comparisons with the *p*-values listed that are close to being significant for DMAP56 at 12, 24, and 48 h. Pairwise comparisons are shown for DMAP56 (top) and DMAP52 (bottom) vs. the control cells for 24 h (lower line): * *p* ≤ 0.05. Results represent means (*n* = 3) ± standard errors of the mean.

**Figure 4 vaccines-11-01085-f004:**
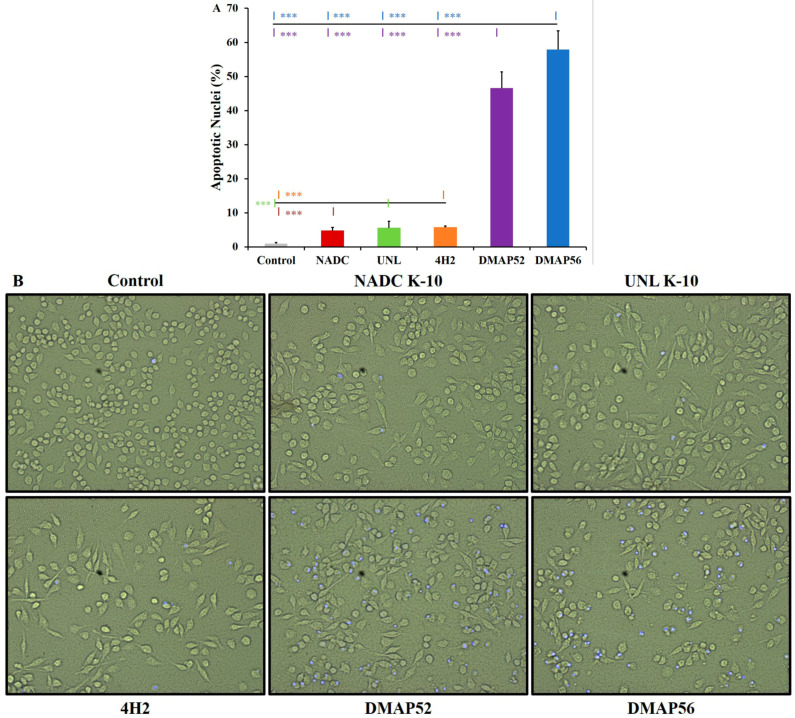
Determination of apoptosis using a rapid staining assay. RAW 264.7 uninfected control (gray) and MAP-infected cells were stained with DAPI at 6 h post-infection: (**A**) the percent of apoptotic nuclei is displayed. The asterisk color represents the strain to which pairwise comparisons were statistically significant: *** *p* ≤ 0.001. Pairwise comparisons are presented for DMAP56 (top) and DMAP52 (bottom) vs. the control, wild types, and 4H2 (upper line). The lower line shows pairwise comparisons between 4H2 (top), UNL K-10 (inline), and NADC K-10 (bottom) vs. the uninfected cells. Results represent means (*n* = 3) ± standard errors of the mean; (**B**) phase contrast (left) and fluorescent (right) micrographs for representative samples for the uninfected cells and each strain. These images were merged using Adobe Photoshop Version 24.4.1 (San Jose, CA, USA) and modified for uniformity and clarity.

**Figure 5 vaccines-11-01085-f005:**
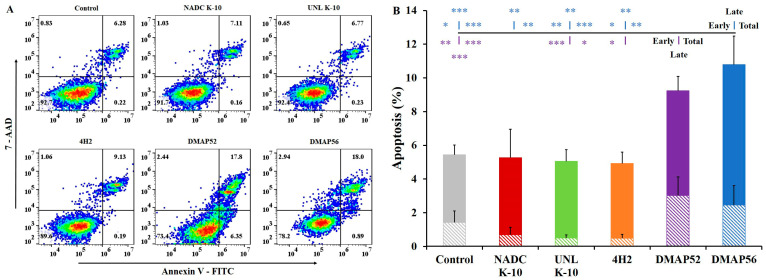
RAW 264.7 uninfected control (gray) and MAP-infected cells were assayed using flow cytometry 6 h post-infection: (**A**) dot plots (FlowJo™ software) for representative samples for the control and each strain are displayed with cell percentiles for each quadrant (see results in Section 3.4 for details); (**B**) the percent of early (diagonal lines), late (solid), and total (early plus late) apoptosis are presented. Pairwise comparisons are shown for DMAP56 (top) and DMAP52 (bottom) for early, late, and total apoptosis vs. the control, wild types, and the transposon mutant. The asterisk color represents the strain to which pairwise comparisons were statistically significant: * *p* ≤ 0.05, ** *p* ≤ 0.01, and *** *p* ≤ 0.001. Results represent means (*n* = 3) ± standard errors of the mean, with varying technical replicates (r) as follows: Control r = 13, NADC K-10 r = 4, UNL K-10 r = 6, 4H2 r = 5, DMAP52 r = 13, and DMAP56 r = 11.

## Data Availability

All data will be available upon request. Please contact rbarletta@unl.edu.

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
