# Peer review of "Mycobacterium avium subsp. paratuberculosis Candidate Vaccine Strains Are Pro-apoptotic in RAW 264.7 Murine Macrophages"

_vaccines, 2023, doi:10.3390/vaccines11061085_

Round 1

Reviewer 1 Report

Comment 1: Attenuation studies of MAP_1152 and MAP_1156 were carried out on Bovine monocyte derived macrophages (Reference 26) whereas the pro-apototic characterization of these mutants in the present study were done on murine macrophages RAW264.7. These are two very different model systems and the results of the present study needs to be correlated with Bovine MDMs data to establish the claim.

Comment 2: TEM studies revealed MAP_1152 and MAP_1156 mutants were significantly elongated in shape with bulged cell wall. This resulted in 10-18 folds lower CFU count of MAP_1152 and MAP_1156 as compared to wild type UNL K-10 (Results-Growth curve and cellular morphology) despite similar OD600 values. This relationship was taken into consideration for MOI calculation for subsequent infection experiments. However no data to show the differential uptake of the wild type and the mutant strains were provided. Difference in cellular morphology has great impact on cellular uptake and enumeration of the same will present a better correlation on the number of bacilli infecting and enhancement of apoptosis.

Comment 3: Several features of apoptosis and necrosis are illustrated in the manuscript. Authors have not checked Caspase activation and ATP availability, a hallmark for apoptosis. Also TEM which is considered gold standard to confirm apaoptosis should have been performed.

Comment 4: Apoptotic cell death can induce inflammation and promote activation of an Immune response desirable of a good vaccine candidate. However not all apoptosis are immunogenic. So for more effective vaccine designing, understanding of the molecules at the nexus of apoptotic death and Innate and Adaptive immunity is desirable. Checking for levels of chemokines such as CCR6 and cytokines like IL-1β, IL-18, Type I Interferons etc will present a better picture on the vaccine candidature of MAP_1152 and MAP_1156 mutants.

Comment 5: Statistical significance not shown in Figure 1.

Author Response

Comment 1: Attenuation studies of MAP_1152 and MAP_1156 were carried out on Bovine monocyte derived macrophages (Reference 26) whereas the pro-apoptotic characterization of these mutants in the present study were done on murine macrophages RAW264.7. These are two very different model systems and the results of the present study needs to be correlated with Bovine MDMs data to establish the claim.

  • We thank the reviewer for all of the important comments. We agree with the reviewer that it would be a valuable confirmation to show that vaccine strains are apoptotic in bovine macrophages as well. We plan to do these experiments in the future. Nonetheless, we would like to point out that two manuscripts have already established a strong correlation between the immunogenic (Adams and Czuprynski, 1994) and apoptotic properties (Wang et al, 2019) of bovine and RAW 264.7 macrophages. Thus, we set up our experiments in RAW 264.7 macrophages to ensure reproducibility of the data as indicated in the manuscript. We have added comments on these studies in both the Introduction and Discussion sections.

Comment 2: TEM studies revealed MAP_1152 and MAP_1156 mutants were significantly elongated in shape with bulged cell wall. This resulted in 10-18 folds lower CFU count of MAP_1152 and MAP_1156 as compared to wild type UNL K-10 (Results-Growth curve and cellular morphology) despite similar OD values. This relationship was taken into consideration for MOI calculation for subsequent infection experiments. However no data to show the differential uptake of the wild type and the mutant strains were provided. Difference in cellular morphology has great impact on cellular uptake and enumeration of the same will present a better correlation on the number of bacilli infecting and enhancement of apoptosis.

  • We agree with the reviewer that this is relatively important issue to clarify. We observed that the uptake of DMAP52 was decreased while DMAP56 was slightly increased in bovine macrophages as stated in the original Reference 26 (Barletta et al., 2019). Moreover, current literature emphasizes shape rather than size as the main determinant of bacterial uptake (Baranov et al., 2021, Frontiers in Immunology, Vol 11). In the case of our strains, the difference in length is approximately 1.4 fold which would not change the uptake significantly since this effect occurs for sizes between 2 - 4 µm (Doshi and Mitragotri, 2010, PLOS One, Vol 5). We have added these references and a comment on this issue in the Discussion. See also response to Reviewer 2 in relation to uptake, intracellular killing and survival.

Comment 3: Several features of apoptosis and necrosis are illustrated in the manuscript. Authors have not checked Caspase activation and ATP availability, a hallmark for apoptosis. Also TEM which is considered gold standard to confirm apoptosis should have been performed.

  • The reviewer should consider that our goal was to determine apoptosis by a rapid procedure (so that mutants can be screened) and not to perform a detailed analysis of apoptosis. Thus, we have not checked for caspase activation or performed TEM studies for apoptosis. Flow cytometry, TEM and caspase activation assays are all considered state of the art, and we used one of them (flow cytometry) to validate the other methods.

Comment 4: Apoptotic cell death can induce inflammation and promote activation of an Immune response desirable of a good vaccine candidate. However not all apoptosis are immunogenic. So for more effective vaccine designing, understanding of the molecules at the nexus of apoptotic death and Innate and Adaptive immunity is desirable. Checking for levels of chemokines such as CCR6 and cytokines like IL-1β, IL-18, Type I Interferons, etc. will present a better picture on the vaccine candidature of MAP_1152 and MAP_1156 mutants.

  • We have added a statement on the immunogenicity based on the cell proliferation assays (original Reference 26; Barletta et al., 2019). As stated in Comment 1, we have also added references indicating the correlation of the immune response and apoptotic processes between bovine and RAW 264.7 macrophages, as shown in the literature.

Comment 5: Statistical significance not shown in Figure 1.

We have added the slope values for OD and CFU for all strains and indicated significant differences where appropriate. We also added the p-values in the text.

Reviewer 2 Report

This article is covering Mycobacterium avium partuberculosis (MAP) as a candidate Vaccine strains in RAW263.7 murine macrophages.

The studies presented in the manuscript were, directed for the development of a model cell culture system to rapidly screen MAP mutant with vaccine potential for apoptosis. Two wild types (transposon mutant and deletion mutant) were selected for testing in murine RAW 264.7 macrophages. The overall goal of the study was determination of the possibility of induction of apoptosis and/or necrosis. The increased level of apoptosis was confirmed in the deletion mutants, which correlated nicely with the attenuated phenotype and immunogenic (characteristic for bovine macrophages) and consequently as very promising and good vaccine candidates. Additionally, 5 very informative figures clearly described the collected data indicative of anticipated results. Particularly the Figure 5 the presenting assay by flow cytometry proved the end results. This will constitute the important goals and novelty of this important paper.

The following suggested changes and recommendations should be introduced before the publication of the manuscript.

1.     Page 5-6, Figure 2 is on two pages and A & B should be on one single page. 

2.     Discussion.  Page 10. Line 367. Insert “thus” before “facilitating” 

3.     Line 388. This paragraph should be separated into “5. Conclusion”

4.     Page 11. Line 415. “Acknowledgments“ “We thank you” Alternatively, insert  “to” before “Joe” or replace “thank” with “thanks” 

The manuscript is of very good quality and urgent importance and is well written and edited in order to meet the standard for the articles published in Vaccines. Thus, I certainly recommend it for publication after the correction of these suggested minor changes. 

Author Response

This article is covering Mycobacterium avium paratuberculosis (MAP) as a candidate Vaccine strains in RAW264.7 murine macrophages.
The studies presented in the manuscript were, directed for the development of a model cell culture system to rapidly screen MAP mutant with vaccine potential for apoptosis. Two wild types (transposon mutant and
deletion mutant) were selected for testing in murine RAW 264.7 macrophages. The overall goal of the study was determination of the possibility of induction of apoptosis and/or necrosis. The increased level of apoptosis
was confirmed in the deletion mutants, which correlated nicely with the attenuated phenotype and immunogenic (characteristic for bovine macrophages) and consequently as very promising and good vaccine candidates. Additionally, 5 very informative figures clearly described the collected data indicative of anticipated results. Particularly the Figure 5 the presenting assay by flow cytometry proved the end results. This will constitute the important goals and novelty of this important paper.
We thank the Reviewer for the insightful and laudatory comments for our manuscript. The Reviewer totally understood the main goal of the manuscript that was to develop and validate a rapid system to screen mutants for apoptotic and necrotic properties.

The following suggested changes and recommendations should be introduced before the publication of the
manuscript.
1. Page 5-6, Figure 2 is on two pages and A & B should be on one single page.

In the revised version (no markups in word), Figure 2 occupies one page. However, the final version prepared by the Journal could have different formatting as they see fit.

2. Discussion. Page 10. Line 367. Insert “thus” before “facilitating”
We added.

3. Line 388. This paragraph should be separated into “5. Conclusion”

We created this new section as requested and removed the words “In summary”.

4. Page 11. Line 415. “Acknowledgments“ “We thank you” Alternatively, insert “to” before “Joe” or replace
“thank” with “thanks”

We did not do this change since his name is You Zhou and he goes by “Joe”. To clarify this better, we added Dr. which we accidentally omitted previously. Thanks.

The manuscript is of very good quality and urgent importance and is well written and edited in order to meet the standard for the articles published in Vaccines. Thus, I certainly recommend it for publication after the
correction of these suggested minor changes.

We thank the Reviewer for his kind comments.

Reviewer 3 Report

In their manuscript, Barletta and coworkers develop an in vitro model to screen for Mycobacterium avium subsp. paratuberculosis (MAP) mutants conditioning pro-apoptotic features after infection of RAW 264.7 murine macrophages. For that purpose, they used five different strains of MAP and three different methods (real time assay, DAPI staining and flow cytometry) to show that apoptosis was followed by secondary necrosis. Overall, the manuscript is well written and presented. However, a few points still require some clarification.

Line 95: Please recapitulate the putative role of the proteins encoded by the genes MAP_1152 and MAP_1156 (the Discussion paragraph is a bit late for that).

Lines 109-111: What are the three knockout mutants? Can the insertion of Tn5367 be considered as a ‘knockout’ since additional DNA was inserted inside the bacterial chromosome? In addition, where is the transposon inserted (the Discussion paragraph is a bit late for that too)?

Line 138: thru

Line 140: How did the authors get rid of the extracellular MAP after infection of RAW 264.7 macrophages?

Line 187-188: Considering the authors’ remark “similar in vitro growth properties in broth culture, a desirable property for vaccine development”, does that mean that DMAP52/56 are not good vaccine candidates since they display a different cell morphology from the WT strains? Having a different cell morphology implies to have different growth properties.

Lines 190-191: Where is the strain 4H2 different? Which time point? Alternatively, what are the slope values?

Figures 3, 4 and 5: Did the authors perform macrophage cell lysis and CFU determination right after infection (one hour?) by the five different MAP strains? Because of the absence of MAP_1152 and MAP_1156 (normally present at the bacterial cell surface) and the subsequent morphological changes (lines 358-359), the observed phenotype may be due to a variable level of phagocytosis. Thus, an increased apoptotic profile in RAW 264.7 macrophages may be the consequence of a higher bacterial load. This is in line with the authors’ comment “in vivo studies in MAP concluded that apoptosis is greater when macrophages contain higher numbers of bacilli” (lines 65-67).

Figure 4B: May it be possible to provide merged pictures?

Lines 346-351: “Inhibition of apoptosis was not observed” in comparison with what? According to the authors, was a limitation due to the use of murine macrophages instead of bovine ones? Otherwise, what is the link with reference 41? Please clarify this paragraph.

Author Response

In their manuscript, Barletta and coworkers develop an in vitro model to screen for Mycobacterium avium subsp. paratuberculosis (MAP) mutants conditioning pro-apoptotic features after infection of RAW 264.7 murine macrophages. For that purpose, they used five different strains of MAP and three different methods (real time assay, DAPI staining and flow cytometry) to show that apoptosis was followed by secondary necrosis.
Overall, the manuscript is well written and presented. However, a few points still require some clarification.

  • We thank the Reviewer for the general and specific comments. Responses are indicated below.

Line 95: Please recapitulate the putative role of the proteins encoded by the genes MAP_1152 and MAP_1156 (the Discussion paragraph is a bit late for that).

  • We moved the details on these proteins to the Introduction as the reviewer suggested.

Lines 109-111: What are the three knockout mutants? Can the insertion of Tn5367 be considered as a ‘knockout’ since additional DNA was inserted inside the bacterial chromosome? In addition, where is the transposon inserted (the Discussion paragraph is a bit late for that too)?

  • Since “knockout” is somewhat undefined as it could be applied to an insertional inactivation or a deletion mutation, we deleted the word “knockout” in this sentence to avoid any potential confusion. We reworded this paragraph to better expand on the mutant descriptions as stated in the original Reference 23 (Rathnaiah et al., 2014). In this way, the manuscript is self-contained and readers that would like more detail can go to this reference. We now specifically describe where the insertion is located on the chromosome in the Materials and Methods section.

Line 138: thru Line 140: How did the authors get rid of the extracellular MAP after infection of RAW 264.7macrophages?

  • For all of the experiments, we added streptomycin and penicillin at 50 ug/ml each to keep the sterility of the media. In addition, streptomycin has the ability to kill extracellular MAP (Zanetti et al., 2006, Annals of Clinical Microbiology and Antimicrobials, Volume 5). These statements were added to the text in Section 2.2. We have modified the methods to include these additions that were left out in the original version. Moreover, we added minor modifications for the DAPI staining process regarding the use of PBS washings that further helps to eliminate extracellular bacteria and is required for the follow up procedures, as was already indicated for the flow cytometry expereiments. For the Real-time assay, the protocol from Promega did not indicate to perform any washes and we relied on the antibiotic to kill extracellular bacteria.

Line 187-188: Considering the authors’ remark “similar in vitro growth properties in broth culture, a desirable property for vaccine development”, does that mean that DMAP52/56 are not good vaccine candidates since they display a different cell morphology from the WT strains? Having a different cell morphology implies to have different growth properties.

  • Based on the OD and CFU slopes, the growth rates (measured in exponentially growing cells) of DMAP52 and DMAP56 are the same as K-10. Thus, having different morphologies as shown in Figure 2, does not result in different growth rates. However, as shown by the CFU curves, the saturation densities of the cultures were different. In that sense, they could be said to have different growth “properties” but do not influence our ability to grow these strains in vitro to carry out animal infections as enough culture will be prepared to compensate for the lower CFU associated with a given OD. We changed “growth properties” and replaced it with “growth rates” as it is what the Figures 1A and 1B are

Lines 190-191: Where is the strain 4H2 different? Which time point? Alternatively, what are the slope values?

  • We have added the Day 0 to 14 slope values during exponential growth (not for an isolated point but a range) for OD and CFU in Figure 1 for all strains and indicated significant differences where appropriate as suggested by this reviewer. Moreover, the slopes of ODs are slightly different, but the slopes based on CFUs are not significantly different. See also response to Reviewer 1 regarding the strain 4H2 that was used more as a control strain rather than a candidate vaccine.

Figures 3, 4 and 5: Did the authors perform macrophage cell lysis and CFU determination right after infection (one hour?) by the five different MAP strains? Because of the absence of MAP_1152 and MAP_1156 (normally present at the bacterial cell surface) and the subsequent morphological changes (lines 358-359), the observed phenotype may be due to a variable level of phagocytosis. Thus, an increased apoptotic profile in RAW 264.7 macrophages may be the consequence of a higher bacterial load. This is in line with the authors’ comment “in vivo studies in MAP concluded that apoptosis is greater when macrophages contain higher numbers of bacilli” (lines 65-67).

  • This question is also related to Comment 2 from Reviewer 1 (please see our reply to this comment as well). We experimented with the Dil dye (1,1'-Dioctadecyl-3,3,3',3'-Tetramethylindocarbocyanine Perchlorate) as used by Kabara and Coussens, 2012. The object of this dye was to both be able to separate bystander from infected cells and assess the number of MAP bacilli at all of the time points. However, we were unable to identify MAP-stained cells due to dye interference in the flow cytometry channels. In our response to Comment 2 by Reviewer 1, we observed that the uptake of DMAP52 was decreased while DMAP56 was slightly increased in bovine macrophages as stated in the original
    Reference 26 (Barletta et al., 2019). Moreover, current literature emphasizes shape rather than size as the main determinant of bacterial uptake (Baranov et al., 2021, Frontiers in Immunology, Vol 11). In addition, the apoptotic properties will be influenced by the total number of bacilli at the time of the assay that will be an interplay of MAP phagocytosis, intracellular killing, survival and eventual replication. We also know that the bacillar numbers are decreased in bovine macrophages (that behave similar to RAW 264.7 cells) for the deletion mutants.
  • Regarding the comment on bacterial load and apoptosis, this was observed in vivo in foamy macrophages and is not relevant to the tissue culture experiments. We have modified the text as follows: Interestingly, in vivo studies in MAP concluded that apoptosis is greater when macrophages contain higher numbers of bacilli, but the significance of these results has not been elucidated [14]. This may represent a later stage in JD as MAP bacilli were found in foamy macrophages and this process would not apply to tissue culture experiments assessing the initial interactions.

Figure 4B: May it be possible to provide merged pictures?

  • Yes, we merged the images and provided the updated pictures.

Lines 346-351: “Inhibition of apoptosis was not observed” in comparison with what? According to the authors, was a limitation due to the use of murine macrophages instead of bovine ones? Otherwise, what is the link with reference 41? Please clarify this paragraph.

  • We meant that inhibition of apoptosis was not observed for the wild type strains as other studies (Kabara and Coussens, 2012) have shown when bystander and infected macrophages were separated. We have re-written the text as follows: However, inhibition of apoptosis was not observed for both wild type strains in our experiments as it was in a previous study that separated bystander from infected cells [13]. The use of a murine macrophage cell line instead of monocyte-derived bovine macrophages is a
    potential reason for not observing apoptosis inhibition by wild type cells, although the omission of cell separation is the most likely explanation for this effect. Moreover, apoptotic processes in bovine macrophages and RAW 264.7 cells are very similar as the same effector (e.g., the programmed cell death protein-4) and regulator (e.g., miR-150) mechanisms are present in both bovine and mouse macrophages [19]. Thus, the RAW 264.7 cell line is an excellent model to study MAP apoptosis, as our results demonstrate that MAP strain pro-apoptotic properties can be assessed in this cell line.

Round 2

Reviewer 3 Report

The authors have addressed all my concerns.

Minor point

The legend of Figure 4B is not up-to-date. Alternatively, the authors can provide three different panels (phase contrast, fluorescence and merged images) for each strain.